# Threats Detection during Human-Computer Interaction in Driver Monitoring Systems

**DOI:** 10.3390/s22062380

**Published:** 2022-03-19

**Authors:** Alexey Kashevnik, Andrew Ponomarev, Nikolay Shilov, Andrey Chechulin

**Affiliations:** St. Petersburg Federal Research Center of the Russian Academy of Sciences (SPC RAS), St. Petersburg Institute for Informatics and Automation of the Russian Academy of Sciences, 199178 St. Petersburg, Russia; ponomarev@iias.spb.su (A.P.); nick@iias.spb.su (N.S.); chechulin@comsec.spb.ru (A.C.)

**Keywords:** intelligent transportation systems, threats detection, smartphone sensors

## Abstract

This paper presents an approach and a case study for threat detection during human–computer interaction, using the example of driver–vehicle interaction. We analyzed a driver monitoring system and identified two types of users: the driver and the operator. The proposed approach detects possible threats for the driver. We present a method for threat detection during human–system interactions that generalizes potential threats, as well as approaches for their detection. The originality of the method is that we frame the problem of threat detection in a holistic way: we build on the driver–ITS system analysis and generalize existing methods for driver state analysis into a threat detection method covering the identified threats. The developed reference model of the operator–computer interaction interface shows how the driver monitoring process is organized, and what information can be processed automatically, and what information related to the driver behavior has to be processed manually. In addition, the interface reference model includes mechanisms for operator behavior monitoring. We present experiments that included 14 drivers, as a case study. The experiments illustrated how the operator monitors and processes the information from the driver monitoring system. Based on the case study, we clarified that when the driver monitoring system detected the threats in the cabin and notified drivers about them, the number of threats was significantly decreased.

## 1. Introduction

Human–computer interaction (HCI) has been an important topic of research and development in recent years. Unfortunately, such interaction creates a lot of vulnerabilities for both sides. On one side, a computer system can expose a human to various threats, while the human can exposure the computer system to various threats. Let us consider the interaction in the example of driver monitoring in an intelligent transportation system (ITS). Such systems can have viruses or some unpredictable behavior that creates vulnerabilities for the driver (human). Alternatively, the driver (human) can be in a fatigued or inattentive state, which creates vulnerabilities for the ITS.

Together with this, a human operator plays an important role in many information and control systems. Therefore, it is crucial to ensure that, on the one hand, the operator is vigilant and is focused on his/her task, and on the other hand, he/she performs actions relevant to the situation (e.g., follows some existing procedure). This confirms the importance of operator behavior monitoring.

We show a general scheme of the driver and operator interaction in the driver monitoring system in Figure 1. An in-cabin driver monitoring system is responsible for the detection of threats caused by a driver in the vehicle cabin. The detected threats are sent to the cloud server that is responsible for server-side processing, collecting, and visualization of the threats in a convenient for the human operator form. On the other hand, the operator is also monitored by the local operator monitoring system for threat detection. The detected threats are sent to the cloud server, where the information is also processed, accumulated, and analyzed.

There are existing research efforts aimed at the analysis of the state of a vehicle driver or of a computer operator [1,2,3]. They are either concentrated on data processing or machine learning model training/development. However, there is no a systematic method covering all stages related to this process and integrating various approaches for the detection of possible threats related to the human state. We cover this gap in the case of a human–computer interaction in a driver monitoring system. The originality of our approach is that we frame the problem of the threat detection in a holistic way: we build on the driver–ITS system analysis performed in an earlier paper [4], and generalize existing methods for driver state analysis into a threat detection method, covering the identified threats. In the paper, we concentrate on the threats related to the human (driver and operator). In our previous paper we proposed a threat classification in the area of intelligent transportation systems [4]. In this paper, we propose a threat detection method that considers how to detect threats in vehicle cabin, as well as a reference model for operator–computer interaction. Based on the proposed method and reference model, we implemented experiments that allowed us to estimate the impact of the driver monitoring system on traffic safety.

The structure of the paper is as follows. In Section 2, we present the related work on the topic of HCI-related threats, and threat identification approaches (in ITS and beyond) in general. We present the threat detection method in Section 3. The reference model of the operator–computer interaction interface is presented in Section 4. Experiments are presented in Section 5. The conclusion summarizes the paper.

## 2. Related Work

This research on threat detection during human–computer interaction touches on three lines of related work. The first line is dedicated to controlling and monitoring the process of human–computer interaction (to ensure that it is adequate to the situation). The second line is aimed on leveraging human–computer interaction elements, as a source of information to detect threats. Finally, as the majority of state-of-the-art methods rely on machine learning and artificial intelligence, the third line of related research is represented by general methodologies, to organize the machine learning process. In this section, we briefly characterize important results in each of these lines of research.

The papers [5,6] propose EYE-on-HCI; an approach and framework for monitoring human–machine interaction during human control of a cyber-physical system (with the example of a nuclear power plant control room). The proposed EYE-on-HCI framework is poised to provide an independent closed-loop validation of human-in-the-loop CPS, by visually gathering data from HCI. Successful data logging of temporal HCI events can be used by an expert supervisory system to correlate real-time plant process data obtained from the plant information system. Finally, EYE can generate cross-validation overview displays and reports for a human supervisor to monitor operator command response in relation to the live control room HCI state. The authors claim that the approach helps to reduce human-in-the-loop errors inherent in feedback control systems and improve overall safety. The proposed framework could be translated to various industrial applications.

A widely used method for identifying and screening vulnerabilities is system network analysis. A network-based model of the system is built, and the effect of denial of each node is propagated through this network, to understand the consequences. Furthermore, the detected vulnerabilities may be ranked and organized using some external criteria to prioritize countermeasures (e.g., [7]). We performed a similar analysis for a vehicle control scenario in our earlier paper [4] and found that abnormal driver states pose severe safety risks, both to the driver and to the whole transportation system. Therefore, a method to identify such driver states needs to be developed.

Most of the modern approaches to analyzing driver state and identifying abnormal behavior are based on machine learning [8,9,10]. There is a large body of knowledge on different machine learning approaches; however, the basic schema of applying machine learning to solve a real-world problem is refined in the MLOps field (the AI domain has some fundamentally different aspects from both software development [11] and data mining [12] and, therefore, requires its own specific process). For example, the paper [12] proposes a process model for the development of machine learning applications, which covers six phases, from defining the scope, to maintaining the deployed machine learning application. The process model expands on CRISP-DM, a data mining process model that enjoys strong industry support but lacks the ability to address machine learning specific tasks. It is an industry and application neutral process model tailored to machine learning applications with a focus on technical tasks for quality assurance.

A study by software teams at Microsoft developing AI-based applications [11] identifies a nine-stage workflow process. This process is based on experiences of developing AI applications (e.g., search and NLP) and data science tools (e.g., application diagnostics and bug reporting).

The paper [13] provides a comprehensive survey of the state of the art in the assurance of ML, i.e., in the generation of evidence that ML is sufficiently safe for its intended use. The survey covers the methods capable of providing such evidence at different stages of the machine learning lifecycle; i.e., of the complex, iterative process that starts with the collection of the data used to train an ML component for a system, and ends with the deployment of that component within the system. 

The presented literature analysis has shown that most of the existing methods of driver analysis are aimed at detecting a particular effect/state. A holistic approach of treating a driver as a part of an intelligent transportation system is missing. At the same time, a driver is an inextricable element of such systems, introducing various vulnerabilities. To fill this gap, this paper proposes a holistic method to detect driver-associated threats in real-time.

Furthermore, emerging process structures for AI-driven solutions suggest several processes that have to be included into the threat detection method, in order to make it effective and reliable: separating model preparation activities, from model application activities, and from background model monitoring processes, aimed at evaluating how a model functions in changing real-life environments (and possibly shifting data distributions).

## 3. Threat Detection Method

The proposed method of threat detection during human–computer interaction in driver monitoring systems is shown in Figure 2. The method integrates findings related to both the driver state analysis and the computer operator state analysis. Both working with a PC or driving are often monotonous processes that require significant attention. As a result, identifying threats such as fatigue, inattention, or irritation can be of high importance. Only analyzing time of work is not applicable, since there are numerous factors affecting the state of the human. In this regard, the method systematizes existing approaches to human state evaluation and applies these, both to the driver and PC operator. This is possible due to the fact that physiological parameters of a human related, for example, to the level of fatigue do not depend on the activity. In fact, the corresponding referenced works analyzed human states in different environments and circumstances. As a result, in this method they can be equally applied to both the driver and PC operator.

The proposed method integrates two main parts: the threat detection sequence, and the supporting loop.

The threat detection sequence (top part, indicated by double lines) consists of the following stages: (i) Capture (represented by ‘Capturing’), (ii) Process/Compute analyzed parameters (represented by ‘Pre-processing’ and ‘Parameter Computing’), (iii) Analyze (represented by ‘Driver State Identifying’).

**Capturing.** At this stage, which source data is captured and how it is captured (method/device) is defined. For example, ECG can be used to capture heart rate, or a camera can be used to capture images (video) of the driver. This stage takes place inside the vehicle. Depending on the computational complexity and type of threat, the other three stages may take place either in a vehicle or in a computing cloud.

**Parameter computing.** Threat identification is usually based on the comparison of certain numeric values to some pre-defined or dynamic thresholds. The numeric values to be compared (the computed parameters) are usually not the data directly captured, so certain data processing is required. This stage is normally presented by a formula. For example, the standard deviation of the intervals of instantaneous heart rate values can be calculated based on the ECG data, or PERCLOS can be calculated if it is known when the driver’s eyes were closed and when opened. However, in many cases (e.g., those where the data is obtained via computer vision), the computed parameter cannot be obtained by a simple formula (there is no a straightforward formula to define if the driver’s eyes in the image are closed or open. For this purpose in some cases data pre-processing is required.

**Pre-processing**. The stage is responsible for converting the source data into a form that can be directly used for the analyzed parameter calculation.

**Driver state identifying.** This is the final stage, where the computed parameters are compared to corresponding thresholds, in order to identify the presence or absence of a threat arising from driver–vehicle interaction.

The supporting loop is aimed at improving the threat detection sequence, via updating the machine learning models used in the pre-processing stage and thresholds used at the analyzing stage. This is done in a manner similar to the classical machine learning pipeline.

Building machine learning models requires datasets that contain information in the same form as used in the capturing stage. The dataset is usually split into three subsets: training set (used for training machine learning models and identifying threshold values), validation set (used for validating machine learning models and threshold values), and test set (used for evaluating the quality of the trained models and threshold values). Both validation and test sets are in some ways used to evaluate the quality of the model. The difference, however, is that the validation set is also used to empirically choose the structure of the model, the architecture of the neural network, learning hyperparameters, etc. In other words, it can be used to fully specify a learning-based solution of the problem. The test set is used only to evaluate the quality of the fully-specified model on unseen data.

In machine learning, the validation stage is usually integrated with selecting hyperparameters of machine learning models. It can be considered as an intermediate test, which is used to check which of the developed/trained alternative ML models is better. This means that the validation set is available to the model developers. On the contrary, the test set is used only to check if the final selected model achieves the desired quality, and normally it is available to the quality assurance department/team, and to the developers.

A supporting loop is required, since during the driver monitoring system operation, new data is accumulated that can be used to improve the machine learning models and threshold values, increasing the accuracy of the threat detection.

Table 1 represents the technologies underlying the stages of the threat detection sequence of the developed method, collected via the literature analysis. The first column identifies the type of the threat (defined in our previous paper [4]), and the other columns correspond to the threat detection sequence stages. Three types of threats are considered: fatigue, inattention, and irritation.

The pre-processing is presented by the most popular techniques. However, in some works, other computer vision technologies can be found. A more detailed review of computer vision approaches was presented in [14].

Since this paper is not aimed at presenting a comprehensive state-of-the-art review of all human state identification approaches, the thresholds presented in the last column are also examples from frequently referenced works. Obviously, all of the technologies presented here could be widely used by drivers; however, we consider the technical possibility of driver state identification.

Irritation detection was proposed in [4]. However, it is still under research, and no particular technologies can be mentioned.

**Table 1 sensors-22-02380-t001:** Technologies for stages of the threat detection method.

Threat	Capturing		Pre-Processing	Parameter Computing	Driver State Identifying
	What	How			
Fatigue	Heart rate	ECG (electrocardiogram)	-	Heart rate variability (HRV) evaluated as standard deviation of the intervals of instantaneous heart rate values (SDNN)	Fatigue is detected if SDNN < 141 +/− 39 ms [9].
		fNIRS (near-infrared functional spectroscopy)	-	Oxygenated hemoglobin HbO2	Fatigue is detected if HbO2 > 2 [15]
	Muscle Fatigue	EMG (electromyography)	-	Peak coefficient of the EMG signal $F_{c}=\frac{A}{x_{rms}}$	Fatigue is detected if Fc > 0.15 [16]
	Macroscopic activity of the surface layer of the brain	EEG	-	Specific bursts in the alpha rhythm	Fatigue is detected if specific bursts are present [17]
	Eyes	Camera	Neural networks/Haar cascades	Blinking frequency (Vb)	Fatigue is detected if Vb > 13 times/minute
				PERCLOS (closing time of the eyelids by more than 80%)	Fatigue is detected if the PERCLOS > 28% of the time within one minute [9].
			Neural networks	ELDC (distance between the eyelids)	Fatigue is detected if ELDC > 0.5 [18]
		EOG (electrooculography) sensor	-	Voltage U	Fatigue is detected if U > 50 µV [9]
	Mouth	Camera	Neural networks	Mouth PERCLOS (closing time of the mouth by more than 50%)	Fatigue is detected if the mouth PERCLOS < 30% [9,11]
	Face	Camera	Neural networks	Skin temperature	Fatigue is detected if skin temperature drops by 0.1 °C [9]
	Body	IR Thermometer	-		
	Body	Camera	Neural networks	Breath rate (Tbr)	Fatigue is detected if Tbr < 16 times/min [19]
	Car dynamics	GPS	CatBoost	No specific parameter or threshold. The machine learning classification model identifies the presence/absence of the treat. [10,20]	
Inattention	Face/Head	Camera	Neural networks/Haar cascades	Driver head’s Euler angles (yaw, pitch, roll) detection (RMAX)	Inattention is detected if RMAX ≥ 15° for longer than 2 s [21]
	Eyes	Camera	Neural networks	View direction (RMAX)	
	Driver	Camera	Neural networks	Presence of the pre-defined objects (food/drink, mobile phone, cigarette)	Inattention is detected if a pre-defined object is present for X seconds [22].
Irritation	Noise	Microphone	To be researched	Noise level	To be researched [4]
	Talking	Microphone	To be researched	Time of talking	To be researched [4]
	Irritating sounds	Microphone	To be researched	e.g., repeating noise	To be researched [4]

## 4. Reference Model of Operator–Computer Interaction Interface

We propose a reference model of the operator–computer interaction interface that describes the main processes that the interface supports (see Figure 3). The presented processes allow the operator to automate the processes of driver monitoring (see the driver list and trip processes). To monitor the fatigue state of the operator, we present the operator fatigue detection process, which monitors the operator fatigue level using an RGB camera and alerts him/her, as well other operators, to prevent dangerous situations.

The driver list process allows the operator to see all important information about the drivers the operator should monitor. This information includes driver name, vehicle type, vehicle number, driver status related to traffic violations (detected by driver monitoring system [21]), number of trips, distance traveled, last trip start time, amount of threats detected for the driver, and current status, which is calculated based on the detected threats for the last few minutes. The information provides the operator with the context that characterizes each driver and helps him/her to identify which driver should be monitored more carefully.

Trip process presents the following information to the operator: vehicle route, threats list detected by the driver monitoring system, in-cabin and outside videos related to each threat, and the possibility to mark each threat as wrong or correct, as well as to mark videos applicable for retraining dataset formation. Vehicle route is a graphical visualization on the map of the GPS/GLONASS data (telemetry) acquired from the vehicle in a map. Vehicle route is a convenient representation of the threats detected by the driver monitoring system. We propose to show each identified threat on the route. Therefore, in this case, the operator easily understands where the threat has been detected. Moreover, we propose to supplement each threat with in-cabin and outside videos, to help the operator to understand the situation when the threat was registered. In-cabin and outside videos are 20-s videos that include the situation when the threat was registered. Together with the vehicle route information the operator has the possibility to analyze the in-cabin as well as outside videos recorded during the whole trip (video registration). In the scope of the proposed reference model, we implemented the possibility for the operator to perform actions (mark every threat with labels) with detected threats.

We also propose actions for the detected threats, for driver monitoring system enhancement. The operator has the ability to mark threats as rightly or wrongly detected, as well as to add undetected threats. For the wrongly detected, as well as undetected, threats we provide the operator with an interface that enables him/her to mark images applicable for machine learning model training/fine-tuning (see Section 3). Together with each image, the system stores metainformation that characterizes the threat the image belongs to, as well as parameters related to the image.

In addition, we propose an operator fatigue detection process to monitor his/her state of fatigue. If the operator is tired, his/her productiveness is reduced and the quality of work is decreased. As such, in this case, his/her involvement in the driver monitoring is not productive.

## 5. Case Study

To test the proposed method and reference model, we developed a prototype for threat detection for vehicle driver and a dispatcher interface based on the reference model presented in Section 4. The right side of Figure 4 shows a vehicle route and detected threats. The left side of Figure 4 shows (1) some important information about the trip, including the number of threats detected and such characteristics as overall trip distance, average speed, duration, maximum speed, and maximum acceleration; (2) information about the selected threat in the map. In the presented example we show the inattention threat (see Table 1) that is detected, based on the driver’s head Euler angles [23,24].

The presented threat is accompanied by two videos (in-cabin and outside) that help the operator to understand if the threat was detected correctly or not.

Figure 5 shows an example of a prototype implemented for dataset retraining. The operator has possibilities to choose an image for retraining of the machine learning model. The operator chooses the appropriate images that are extracted automatically from the video. We propose taking five random images from the video sequence. The operator chooses the image and annotates it with additional information (see Figure 6).

Figure 6 shows an example of threat rejection. If the operator decides to reject the detected threat, he/she specifies the reason in the system.

Based on the presented method and reference model, we conducted in-the-wild experiments with 14 drivers and several operators and estimated our driver monitoring system (presented earlier in papers). The main purpose of the experiments was to evaluate the proposed method and estimate how the driver behavior changes if he/she receives notifications about the detected threats. During the experiment, drivers used their cars with the built in driver monitoring system developed based on the proposed method. Using the driver monitoring system, all detected threats were accumulated in the cloud server, and operators checked the videos to accept or reject them. We implemented analysis of the data obtained, based on the experiments that are shown in Table 2. The table shows the results of the experiments for participants that drove for more than 100 km. We conducted experiments in passive and active modes. Passive mode means that the driver monitoring systems detects threats and sends them to the cloud server, but does not inform the driver about them. Active mode means that the system notifies the driver about the detected threats. We show the distance of the overall driver trips, the amount of detected threats, threat frequency for 10 km, and frequency change when the system switched from passive to active mode during the experiment. Based on the experiments, we obtained positive results, showing that active mode significantly decreased the number of threats (e.g., for driver 1 and driver 14, it was more than 50%) compared to the passive mode.

## 6. Conclusions

This paper proposes an approach and underlying method for detecting threats caused by the drivers and operators of an intelligent transportation system. The threats include fatigue, inattention, and irritation. The method integrates a threat identification sequence consisting of ‘capturing’, ‘pre-processing’, ‘parameter computing’, and ‘driver state identifying’ stages; as well as the supporting loop, aimed at the fine-tuning of underlying machine learning models, thresholds, and other parameters used in the threat identification sequence stages. The method was implemented in a prototype driver monitoring system that enables both driver and operator monitoring, although most attention was devoted to driver monitoring. The prototype does not only implement the threat identification sequence, but also a supporting loop, providing operators with instruments to mark ill-identified treats and to extend the training datasets to improve detection in the future. The carried out experiments showed that the identification of threats by the developed method and informing the driver about them significantly reduced the number of threats regarding the driver. The main limitation of the experiments was that we evaluated only the part of the method related to driver monitoring. At the moment, the operator monitoring part is not ready for evaluation; which is our future work. In addition, future work is planning to address two main directions. The first direction is related to finalizing the mentioned dataset and making it publicly available. It will contain, not only videos, but also recordings of various physiological parameters, such as heart rate or breathing rate. The other direction is related to the development of approaches and models for detecting threats related to the distraction of a driver or computer operator. Currently, they are only mentioned in the presented method as ‘to be developed’. Based on the current findings and the collected dataset, we plan to develop corresponding machine learning models and perform experiments to analyze their efficiency.

## Figures and Tables

**Figure 1 sensors-22-02380-f001:**
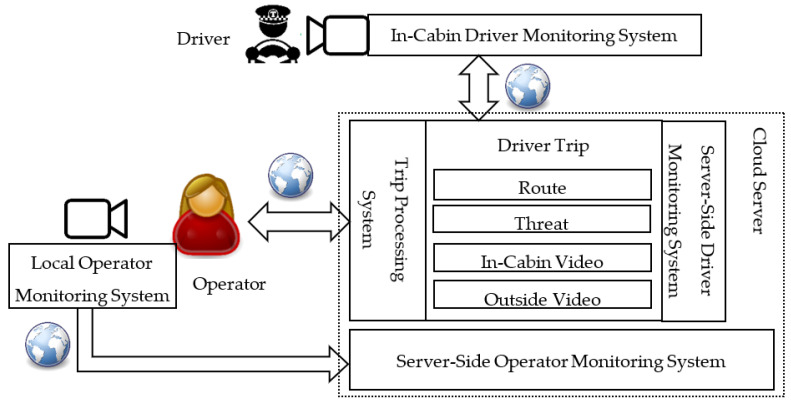
General scheme of the driver and operator interaction in the driver monitoring system.

**Figure 2 sensors-22-02380-f002:**
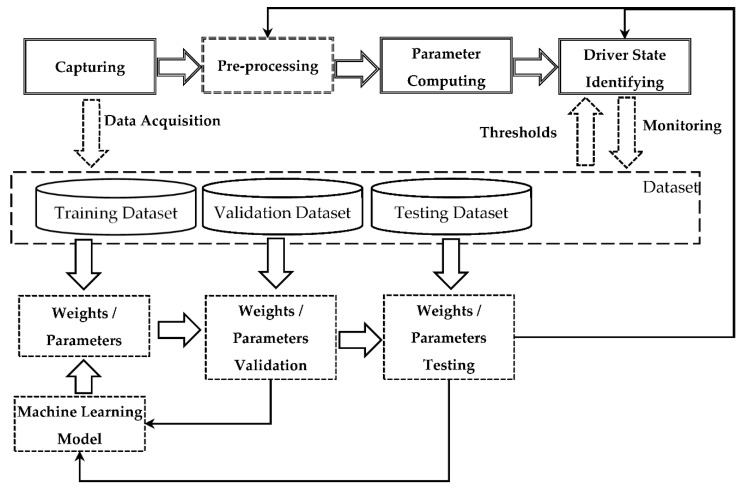
Proposed threat detection method.

**Figure 3 sensors-22-02380-f003:**
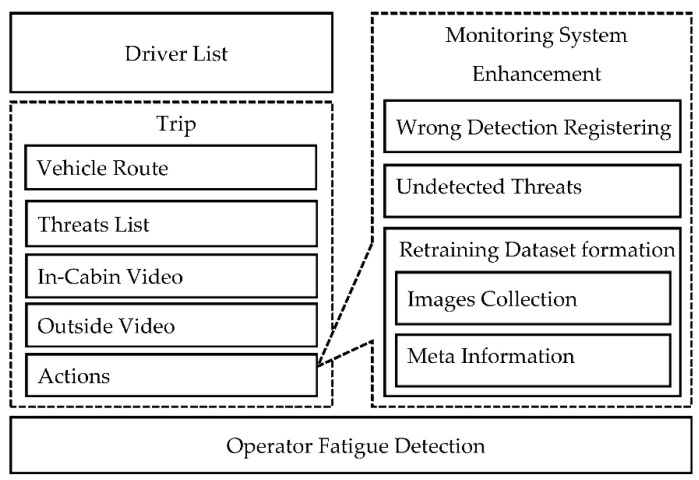
Reference model of operator–computer interaction interface.

**Figure 4 sensors-22-02380-f004:**
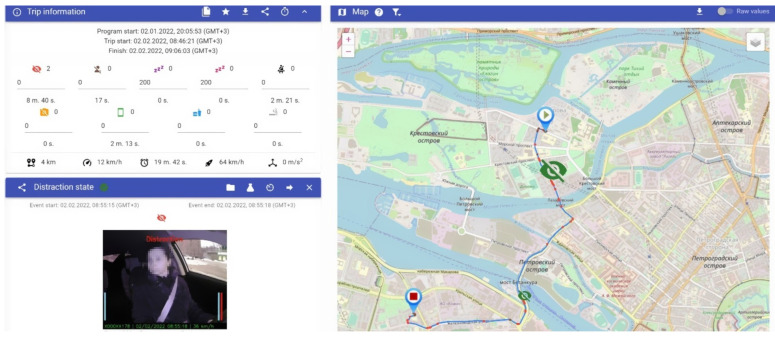
Screenshot example: vehicle route, threat lists, in-cabin, and outside videos.

**Figure 5 sensors-22-02380-f005:**
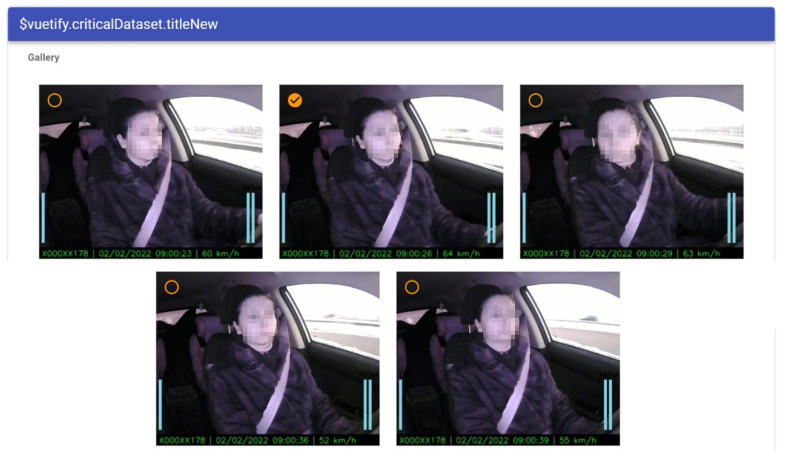
Screenshot example: choosing images for dataset retraining.

**Figure 6 sensors-22-02380-f006:**
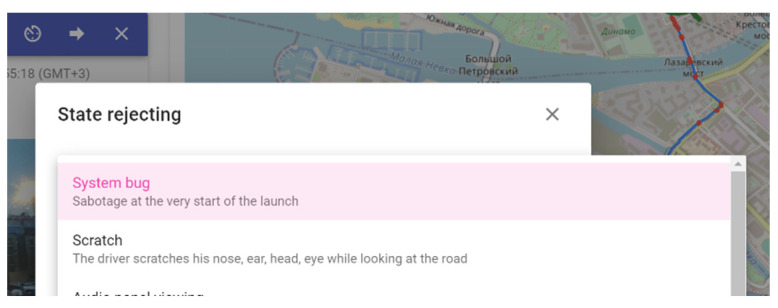
Screenshot example: rejecting of the detected threat by a dispatcher.

**Table 2 sensors-22-02380-t002:** In-the-wild experiments.

Drivers	Trips, km		Threats		Threat Frequency (Pc. on 10 km)		Frequency Change	
	Passive Mode	Active Mode	Passive Mode	Active Mode	Passive Mode	Active Mode	Pc. on 10 km	%
Driver 1	64	49	86	25	13.4	5.1	−8.3	−62.0
Driver 6	562	440	438	190	7.8	4.3	−3.5	−44.6
Driver 7	605	253	424	168	7.0	6.6	−0.4	−5.3
Driver 13	243	147	292	20	12.0	1.4	−10.7	−88.7
Driver 14	220	250	109	59	5.0	2.4	−2.6	−52.4

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
