# Peer review of "Threats Detection during Human-Computer Interaction in Driver Monitoring Systems"

_sensors, 2022, doi:10.3390/s22062380_

Round 1
Reviewer 1 Report
This paper presents an approach for threats detection during human-computer interaction in driver monitoring systems. The proposed paper suggests a novel concept with significant importance. This article has an interesting idea and its basic concepts are well described. But it needs a fundamental change as follows:
- The abstract must be a concise yet comprehensive reflection of what is in your paper. Please modify the abstract according to the motivation, description, results, and conclusion parts.
- What is the motivation of the proposed method? What is the motivation of the present technique? Please add this detailed description in section 1.
- In the introduction part, please consider a review of the existing literature and show what the originality of your work is.
- The study lacks a clear comparison between the submitted paper and the more relevant literature contributions, which should highlight the main advantages of the current submission.
- Add the future work for the proposed technique in the Conclusion.
Author Response
Dear reviewer,
Thank you for the valuable comments. Please see to our detailed answers in the attached PDF file.
Authors.

Reviewer 2 Report
This paper aims at detecting abnormal behaviors of drivers during human-computer interactions, e.g., fatigues, inattention, and irritation, and the authors developed a vision-based monitor system that can be used to detect abnormal behaviors. 14 drivers are invited to test the proposed system. In general, the authors have completed the work, while the innovations and more details of tests should be explained. (1) There are many studies working on a similar theme as the authors presented in Section 3. Those studies used various sensors to detect fatigue, inattention, and irritation. Thus, what are the innovation of this study? (2) The authors mentioned that the Human-computer interaction creates a lot of vulnerability and threats to the human at the beginning of the manuscript. However, the authors’ statements are lack evidence and misleading, which hindered the understanding of this manuscript. Specifically, Would the proposed method identify the fatigue due to the high workload of interaction with the computer? (3) A formal definition of HCI-threats would be needed. (4) A summary of existing techniques would be needed to explain the innovation of this manuscript. (5) The details of the experiment are needed, for instance, the purpose, process, analysis, positive/negative results, and limitations of the experimentsAuthor Response
Dear reviewer,
Thank you for the valuable comments. Please see to our detailed answers in the attached PDF file.
Authors.

Reviewer 3 Report
The paper outlines a method for detecting risks during driver-vehicle interactions, as well as a case example. The driver monitoring system was examined, and two categories of users were identified: the driver and the operator. The suggested method detects potential driver and operator hazards such as weariness, inattention, and irritability. It describe a method for detecting risks during human-system interactions that generalises possible threats and detection approaches. This work is not technically sound to publish in this reputed journal.
- details experimental design is missed.
- no comparison has been provided with state art of methods in the area.
- Data description is missed.
- Poor writing style as 'The paper [8] reports.........................'
Author Response

(The authors gave the same response as above.)

Round 2
Reviewer 2 Report
I have no further comments on the current manuscript.
Reviewer 3 Report
Authors have addressed all suggested revisions/concerns. Manuscript is acceptable in its present form.